# Hepatitis E Virus Seroprevalence and Associated Risk Factors in Apparently Healthy Individuals from Osun State, Nigeria

**DOI:** 10.3390/pathogens9050392

**Published:** 2020-05-20

**Authors:** Folakemi Abiodun Osundare, Patrycja Klink, Catharina Majer, Olusola Aanuoluwapo Akanbi, Bo Wang, Mirko Faber, Dominik Harms, C.-Thomas Bock, Oladele Oluyinka Opaleye

**Affiliations:** 1Department of Medical Microbiology and Parasitology, Ladoke Akintola University of Technology, Osogbo PMB 4400, Nigeria; 777folasegun@gmail.com (F.A.O.); AkanbiO@rki.de (O.A.A.); ooopaleye@lautech.edu.ng (O.O.O.); 2Science Laboratory Department, Federal Polytechnic Ede, Ede 232101, Nigeria; 3Department Infectious Diseases, Viral Gastroenteritis and Hepatitis Pathogens and Enteroviruses, Robert Koch Institute, German Ministry of Health, 13353 Berlin, Germany; klinkp@rki.de (P.K.); catharina.majer@pei.de (C.M.); bowang@vt.edu (B.W.); HarmsD@rki.de (D.H.); 4Host-Pathogen Interactions, Paul-Ehrlich-Institut, 63225 Langen, Germany; 5Department of Biomedical Sciences and Pathobiology, Virginia-Maryland College of Veterinary Medicine, Virginia Polytechnic Institute and State University, Blacksburg, VA 24061, USA; 6Gastrointestinal Infections, Zoonoses and Tropical Infections Unit, Department for Infectious Disease Epidemiology, Robert Koch Institute, German Ministry of Health, 13353 Berlin, Germany; FaberM@rki.de; 7Institute of Tropical Medicine, University of Tuebingen, 72071 Tuebingen, Germany

**Keywords:** hepatitis E virus, seroprevalence, ELISA, PCR, risk factors

## Abstract

Hepatitis E virus (HEV) infection is a major public health concern in low-income countries, yet incidence and prevalence estimates are often lacking. Serum (*n* = 653) and faecal (*n* = 150) samples were collected from apparently healthy individuals using convenience sampling technique in six communities (Ore, Oke-Osun, Osogbo, Ede, Esa-Odo, and Iperindo) from Osun State, Nigeria. Serum samples were analysed for total anti-HEV IgG/IgM and anti-HEV IgM using commercially available HEV ELISA kits. Total anti-HEV positive serum and all stool samples were analysed for HEV RNA by RT-PCR. Overall, 15.0% (*n* = 98/653) and 3.8% (*n* = 25/653) of the serum samples were positive for anti-HEV total and IgM antibodies, respectively. Total anti-HEV and IgM in Ore, Oke-Osun, Osogbo, Ede, Esa-Odo, and Iperindo was 21.0% (*n* = 13/62) and 3.2% (*n* = 2/62), 19.4% (*n* = 20/103) and 6.8% (*n* = 7/103), 11.4% (*n* = 12/105) and 2.9% (*n* = 3/105), 8.0% (*n* = 16/199) and 1.5% (*n* = 3/199), 22.0% (*n* = 22/100) and 10.0% (*n* = 10/100), and 17.9% (*n* = 15/84) and 0.0% (*n* = 0/84), respectively. All samples (stool and serum) were HEV RNA negative. Anti-HEV seroprevalence was associated with rural location, increasing age, alcohol consumption, and rearing of animals. This study demonstrated a high anti-HEV seroprevalence in Osun State, indicating the need to implement surveillance and asses the hepatitis E burden in Nigeria.

## 1. Introduction

Hepatitis E virus (HEV), a single-stranded RNA virus classified in the genus *Hepevirus*, family *Hepeviridae,* is the most common cause of acute hepatitis in humans worldwide, [1]. Isolates of HEV infecting humans belong to the *Orthohepevirus A* species, which so far, consists of eight genotypes based on the phylogeny of the entire viral genomes (HEV-1-HEV-8) [2]. Four genotypes (HEV-1–HEV-4) are known to infect humans, although recently, infection of humans with HEV-7 has also been reported [1,3]. Genotypes 1 and 2 are exclusively found in humans and are mainly associated with outbreaks in low-income countries due to contaminated drinking water. Genotypes 3 and 4 infect humans and various animal species and cause sporadic infections via zoonotic transmission through the ingestions of undercooked animal meat as well as direct contact with infected animal reservoirs [1]. HEV genotypes 1 and 2 are endemic in most low-income countries [4]. A recent HEV genotype 1 and 2 outbreak in Northeast Nigeria with 1815 suspected cases, including eight infection-associated deaths, shows that HEV is of public health concern in Nigeria [5,6]. However, little is known about the seroprevalence and distribution of HEV, as well as risk factors contributing to HEV infection in Nigeria. Existing studies report an anti-HEV seroprevalence ranging from 7% to 93% in different regions of the country [7,8,9,10,11] and genotypes 1 and 2 have been detected in human samples in Nigeria [5,12,13]. In Nigeria, social amenities are still inadequately distributed, such as access to health care and potable water being more concentrated in urban communities, thus resulting in poor standards of living in rural areas of the country which may facilitate infections with HEV. Osun State is a representative of rural and urban regions. Therefore, the analysis of serum samples from apparently healthy individuals from six different communities in Osun State will further contribute to the understanding of the distribution and dynamics of circulating HEV variants and will help to identify risk factors for HEV infection in Nigeria.

## 2. Results

### 2.1. Sample and Participant Characteristics

A total of 653 apparently healthy individuals from six communities in Osun State were included in this study. Most study participants were from Ede community (30.5%; *n* = 199/653), followed by individuals from Osogbo (16.1%; *n* = 105/653), Oke-Osun (15.8%; *n* = 103/653), Esa-Odo (15.3%; *n* = 100/653), Iperindo (12.9%; *n* = 84/653), and Ore (9.5%; *n* = 62/653).

Of the 653 participants, 54.5% were female (*n* = 356/653) and 45.5% were male (*n* = 297/653). The median age was 35 years (IQR: 24–50 years) with the age group 20–29 years being the most (27.0%; *n* = 176/653) and the age group 0–9 years being the least represented (3.4%; *n* = 22/653) (Table 1).

### 2.2. Anti-HEV Seroprevalence and Associated Risk Factors

Of the 653 study participants, 15.0% (*n* = 98/653; 95% CI 12.5–18.0%) were tested positive for anti-HEV total antibodies. Of these, 25.5% (*n* = 25/98; 95% CI 17.2–35.3%) also tested positive for anti-HEV IgM (in total 3.8%; *n* = 25/653; 95% CI 2.6–5.6%). The seroprevalence of anti-HEV antibodies was slightly higher in male than in female for both anti-HEV total antibodies (17.9%; 95% CI 13.7–22.7% versus 12.6%; 95% CI 9.4–16.6%;) and anti-HEV IgM (5.1%; 95%CI 2.9–8.2% versus 2.8%; 95% CI 1.4–5.1%) (Table 1). However, these results were not statistically significant.

Anti-HEV seropositivity increased significantly with age (Table 1). IgG peaked in individuals 50–59 years of age (31.5% (*n* = 23/73; 95% CI 21–43%), while the age group of 60–69 years showed the highest levels of anti-HEV IgM with 11.9% (*n* = 7/59; 95% CI 4.9–22.9%). Overall, anti-HEV total antibody seroprevalence increased with rurality (Esa-Odo (rural) > Ore (rural) > Okeosun (semi-urban) > Iperindo (rural) > Osogbo (urban) > Ede (urban) (Table 2). The highest anti-HEV total antibody seroprevalence was observed in Esa-Odo community with 22.0% (*n* = 22/100; 95% CI 14.3–31.4%), while the lowest seroprevalence was found in Ede community with 8.0% (*n* = 17/199; 95% CI 5.1–13.3%). The frequency of being anti-HEV total antibody positive was nearly two times higher for individuals living in rural regions compared to individuals living in urban regions (aOR = 1.8; 95% CI 1.1–3.0; *p* < 0.05) (Table 2). The highest anti-HEV IgM seroprevalence was detected in individuals from Esa-Odo (rural community) (10%; *n* = 10/100; 95% CI 4.9–17.6%). In Iperindo, which is also a rural community, none of the individuals tested positive for anti-HEV IgM.

A high anti-HEV seroprevalence was observed in individuals rearing animals with 20.9% for anti-HEV total (*n* = 57/273; 95% CI 16.2–26.2%) and 7.0% (*n* = 19/273; 95% CI 4.4–10.7%) for anti-HEV IgM antibodies, compared to individuals without close contact to farm animals (5.8% (*n* = 8/139; 95% CI 2.5–11.0%) and 1.4% (*n* = 2/139; 95% CI 0.2–5.1%)), respectively. Statistical analysis showed that individuals rearing animals have a three times higher risk of being anti-HEV total antibody positive (aOR = 3.0; 95% CI 1.3–6.7; *p* < 0.01) in comparison to individuals not rearing animals.

In addition, alcohol intake had a statistically significant association with anti-HEV total and IgM antibodies (aOR = 2.4; 95% CI 1.3–4.4; *p* < 0.01 and aOR = 2.8; 95% CI 1.0–7.7; *p* < 0.05, respectively) (Table 2). Other risk factors (like blood transfusion history, toilet type used, drug abuse, water source, and house types) did not have a statistically significant association with anti-HEV total or IgM antibodies (Table 2).

Besides age, the multivariate logistic regression final model showed that alcohol consumption (aOR = 2.7; 95% CI 1.3–5.3; *p* = 0.006) and rearing animals (OR = 3.2; 95% CI 1.4–7.3; *p* = 0.005) were the only variables associated with anti-HEV total antibodies. While Esa-Odo (OR = 6.9; 95% CI 1.4–35.0; *p* = 0.02) and Oke-Osun (OR = 5.1; 95% CI 1.0–26.3; *p* = 0.051) among the study locations and alcohol consumption (OR = 3.8; 95% CI 1.3–11.2; *p* = 0.016) were the variables significantly associated with anti-HEV IgM as shown in Table 3.

### 2.3. PCR Detection of HEV

All stool samples and samples positive for anti-HEV total antibodies were subjected to PCR analyses for the detection of HEV RNA. In none of the samples HEV RNA was detected.

## 3. Discussion

This study depicts the distribution of HEV infections in Osun State, Nigeria. The overall anti-HEV total antibody seroprevalence of the selected communities (15.0%) in this study is comparable to the recently described seroprevalence of 13.4% observed in Ekiti State Nigeria [7]. This comparison may be due to similar prevailing conditions such as shared cultural values, similar socioeconomic status and the same geographical region as both Ekiti and Osun States are in the Southwest of Nigeria. Further studies from Nigeria report higher anti-HEV seroprevalences than observed in this study. Ola et al. [10] report an anti-HEV seroprevalence of 44% among health workers and 93% among non-health workers in Ibadan, Oyo State, Southwest Nigeria. In another study by Junaid et al. [9], a seroprevalence of 47.9% was recorded in apparently healthy individuals in Plateau State, Northcentral Nigeria. Smaller sample sizes used by Ola et al. [10] and Junaid et al. [9] may be a probable factor for the discrepancy observed with this study.

In comparison to other African countries, the anti-HEV total antibody seroprevalence observed in this study is higher than the seroprevalence of 5.4% and 4.6% reported among blood donors in Tunisia [14] and Ghana [15], respectively. However, the anti-HEV total antibody seroprevalence in this study is lower than the seroprevalence of 42% in Zambia [16] in apparently healthy individuals. The differences in seroprevalences may be due to the different assays used or various factors fuelling HEV transmission in the different geographical areas.

The same ELISA kits (MP Diagnostic) were used in China [17], US [18], and this study; the observed anti-HEV total antibody seroprevalences were 23.46%, 7.7%, and 15.0%, respectively. However, different HEV genotypes are associated with each geographical region and different factors fuel their transmission. Genotypes 1 and 4 are predominant in China [17] while genotype 3 is predominant in the US [19,20] and in Africa genotypes 1 and 2 are predominant [4]. Different routes of transmission of these HEV genotypes but also differences in the analysed study populations may account for the differences observed in the seroprevalences.

The anti-HEV IgM seroprevalence observed in this study was high (3.8%). In other parts of Nigeria, only a few studies exist. However, in these studies, the anti-HEV IgM seroprevalences reported were low. Anti-HEV IgM seroprevalence was 0.9% among different populations including apparently healthy individuals (anti-HEV IgM seroprevalence = 0%) from Plateau State [9]. Additionally, in community dwellers from Oyo and Anambra State, 0% anti-HEV IgM seroprevalence was reported [21]. However, Junaid et al. [9] observed an anti-HEV IgM seroprevalence of 4.2% in animal handlers, which is consistent with the seroprevalence of 7% observed in individuals rearing animals in this study. The high anti -HEV IgM seroprevalence observed in this study may imply ongoing transmission of HEV in the State.

One reason for the low anti-HEV IgM prevalence in comparison with anti- HEV IgG prevalences in all of these studies might be that, as a marker of acute infection, the persistence of anti-HEV IgM antibodies is known to be short-lived declining to baseline levels after 3–12 months [1]. However, a longer persistence of anti-HEV IgM (up till 34 months) has been reported with some assays and furthermore varying analytical performance of different assays has been shown, which should be taken into account in seroprevalence studies [22,23]. All the anti-HEV IgM positive samples and all the stool samples in this study were negative for HEV RNA, which is probably due to the short diagnostic window of HEV RNA as viremia usually lasts for only between 4 and 6 weeks after infection [1]. Another reason for the negative HEV genome detection in our cohort could be that the HEV viral load of the samples was lower than the sensitivity for ORF 1 nested and ORF 2 semi-nested PCRs (1 × 10^4^ copies/mL (8.4 × 10^3^ IU/mL) and 1 × 10^3^ copies/mL (840 IU/mL), respectively). Hence false negative results cannot be completely ruled out.

The ELISA assay used in this study was the MP Diagnostics HEV ELISA 4.0 (total antibodies) and MP Diagnostics HEV IgM ELISA 3.0 (IgM) for the analysis of the seroprevalences although Wantai IgG ELISA is a more commonly used assay worldwide. However, in several studies the MP anti-HEV total antibodies ELISA has been reported to be comparable or even more robust than the Wantai IgG ELISA in terms of analytical performance [24,25,26]. The MP HEV IgM ELISA has been reported with a specificity of 99.5%. 93% and 84% by the studies of Legrand-Abravanel et al. [27], Drobeniuc et al. [28], and Pas et al. [23], respectively. Wide disparities have been observed in some studies when different HEV IgM ELISAs were used for the same samples [24,25].

### 3.1. Risk Factors for HEV Infection

In this study, a higher anti-HEV seroprevalence (total antibodies) was found in individuals from rural communities compared to individuals from urban communities (20.1% versus 9.2%). The anti-HEV seroprevalences in Ede and Osogbo (urban settlements) were the lowest in the six analysed communities (8.0% and 11.4%, respectively). This could be because of the availability of more social amenities such as accessibility to adequate potable water, better housing system, good sanitary measures, good road network, and healthcare facilities in comparison to the other communities.

In contrast, the seroprevalence of anti-HEV total antibodies in Esa-Odo, Ore and Oke-Osun are similarly high (22.0%, 21.0% and 19.4%, respectively). These communities are mainly rural and agrarian dwellers so the high seroprevalence observed may be due to the farming activities. The use of animal dung as manure, poor hygienic conditions on the farm and consumption of stream water contaminated by human or animal faeces could foster the transmission of HEV. Mixed breeding of pigs with other farm animals or swine faecal contamination of rivers in these communities could have facilitated zoonotic transmission of HEV infection. Generally, unsatisfactory sanitary conditions are more associated with rural areas in low-income countries. Therefore, the above findings from our study corroborate the work of Junaid et al. [9] who reported an association of a higher anti-HEV seroprevalence with rurality. Although the anti-HEV total antibody seroprevalence is similar in Esa-Odo, Ore and Oke-Osun, the anti-HEV IgM seroprevalence in Esa-Odo and Oke-Osun is much higher than that of Ore (10.0% and 6.8% versus 3.2%, respectively). Furthermore, the multivariate analysis revealed that Esa-Odo and Oke-Osun are more associated with anti-HEV IgM than the remaining analysed communities. The lower anti-HEV IgM seroprevalence in Ore could be a result of the impact of an intervention on the spread of Schistosomiasis in Ore. Interventions on Schistosomiasis endemicity in Ore have helped in enlightening the community on the importance of proper hygiene and how to prevent waterborne infections [29] which subsequently might have affected the transmission of other disease agents such as HEV.

### 3.2. Age as a Risk Factor for HEV Infection

The anti-HEV total antibodies seroprevalence in this study was higher in individuals >40 years of age in comparison to younger individuals and was highest in the age range of 50–59 year-olds (31.5%). The highest anti-HEV IgM seroprevalence was found in the age range of 60–69 year-olds. Using age as a continuous independent variable, both total anti-HEV and anti-HEV IgM had a statistically significant association with increasing age. Most studies from Africa report a higher HEV infection risk in young adults [30]. In Nigeria, Oladipo et al. [11] reported a decreasing anti-HEV seropositivity with age in Ogbomoso, Southwest, Nigeria, while Junaid et al. [9] stated that anti-HEV seropositivity increases with age in Plateau State, Northcentral, Nigeria. However, in European countries, where infections are mainly caused by HEV genotype 3, seroprevalence increases with age [31,32]. Therefore, our findings could be due to a lifetime-dependent exposure or could hint to infections with HEV-3 in Nigeria.

### 3.3. Rearing of Animals as a Risk Factor for HEV Infection

Rearing of animals was significantly associated with anti-HEV total antibodies both in the univariate and multivariate analysis, implying that zoonotic transmission might be a potential transmission pathway in this study. A higher seroprevalence of anti-HEV total and IgM antibodies was observed among people rearing animals than people not breeding animals (seroprevalence of anti-HEV total and IgM antibodies = 20.9% and 7.0%, respectively versus 5.8% and 1.4%, respectively). The respondents of the study bred different animals such as goats, dogs, hens, pigs and cows predominantly. El-Tras et al. [33] detected anti-HEV antibodies in cows, goats, and sheep in Egypt in 21.6%, 9.4%, and 4.4% of the animals, respectively. Presence of anti-HEV antibodies has also been documented in Nigerian meat animals such as goats (37.2%), pigs (32.8%), and sheep (10.5%) in the study of Junaid et al. [34]. The presence of HEV in these domesticated animals may serve as a source of infection to the human populace through zoonotic or foodborne transmission. Genotype 3, 4 and 7 have been associated with animals; however, the only genotype documented in animals in Nigeria is genotype 3 which was detected in pigs by Owolodun et al. [35]. Until so far, genotype 3 has not been discovered to circulate in the Nigerian human populace though it has been documented in different parts of the Western world and also rarely on the African continent [36,37]. However, the higher anti-HEV seroprevalence in individuals rearing animals as well as the increasing anti-HEV seroprevalence with increasing age in individuals from this study may imply infections with HEV genotype 3 in Nigeria [38]. Rearing animals may also be a confounding factor; probably people not rearing animals live under better hygienic conditions thus making them less frequently exposed to HEV genotype 1.

### 3.4. Alcohol Consumption as a Risk Factor for HEV Infection

Alcohol consumption had a statistically significant association with anti-HEV total and IgM antibodies. This finding corroborates the findings of Junaid et al. [9], where alcohol consumers had a higher seroprevalence than non-alcohol consumers. Alcohol consumption is likely a co-founding factor of infection with HEV in Osun State as the individuals who consumed alcohol showed a two times higher risk of being anti-HEV-positive. Additionally, a nearly three times higher risk of being anti-HEV IgM-positive was observed in comparison to individuals who stated they did not drink alcohol. Multivariate analysis showed consistently that alcohol consumption is statistically significant with both anti-HEV total antibodies and anti-HEV IgM. Intake of alcohol in this part of the world is usually an outdoor activity often accompanied by consumption of street foods which may have been contaminated. More so, some alcohols are locally brewed and subjected to water of unknown purity. In addition, since abuse of alcohol has a significant impact on the integrity of the liver and immune system of the body; an alcohol user may tend to be more susceptible to infections generally [39].

### 3.5. Limitations of the Study

Limitations of this study were that a convenience sampling technique was used to recruit the study participants in the various communities. All the participants that volunteered to be part of the study were recruited. However, to solve this, the data analyses were adjusted by age and sex. As apparently healthy participants were recruited for the study, individuals with undiagnosed health issues such as HIV or others may have been included in the study. A further limitation of the study is the use of convenience sampling method as it is highly vulnerable to selection bias and therefore might not reflect the general population [40]. The sensitivity for ORF 1 nested and ORF 2 semi-nested RT-PCRs are 1 × 10^4^ copies/mL and 1 × 10^3^ copies/mL, respectively. Thus, it cannot be ruled out that individuals with a lower HEVviral load might have shown false-negative tests. However, a recent study by Lhomme et al. [41] showed that in asymptomatic blood donors a median HEV load of 717 IU/mL was determined and symptomatic individuals showed a median load of 2.82 × 10^5^ IU/mL. Therefore, the PCR detection limit was acceptable for this study analysing apparently healthy individuals showing no hepatitis E-like symptoms.

## 4. Materials and Methods

### 4.1. Study Site

Osun State with a population size of 3.4 million is one of the 36 States of the Federal Republic of Nigeria, as shown in Figure 1:

a. It is in South-West Nigeria and has three senatorial districts (Osun-Central, Osun-East and Osun-West senatorial districts). The study sites are from the three senatorial districts of the State; these include Ede, Esa-Odo, Iperindo, Oke-Osun, Ore and Osogbo, as shown in Figure 1b.

b. Ede extends over Ede-North and Ede-South LGA with a population of 159,866 [42]. Esa-Odo is in Obokun LGA, Osun East senatorial district with a population size of about 2000 Swine were observed to move close to human abodes. Iperindo is in Atakumosa East LGA with a population size of about 10,000. Oke-Osun is a farm settlement area located in the outskirt of Osogbo, Osogbo LGA of Osun-Central senatorial district with a population size of about 1000. The inhabitants are predominantly farmers with activities ranging from crops planting to animal husbandry. Ore is in Odo-Otin LGA, Osun Central senatorial district with a population size of about 2000 [43]. Osogbo is the capital of Osun State; it expands over three local government areas (Olorunda, Osogbo and Egbedore LGAs). Osogbo is well endowed with social amenities as compared to all other communities in the State. The population size is approximately 156,694 [42]. Esa-Odo, Iperindo, and Ore are rural communities with most of the populace being predominantly farmers and petty traders. Ede and Osogbo are urban communities, and major commercial centres. All the study sites have water bodies in proximity.

### 4.2. Sample Collection and Processing

From a total of 653 consenting (apparently healthy) individuals, sera (*n* = 653) and stool samples (*n* = 150) were collected from six communities in Osun State. These communities were carefully chosen to represent the three senatorial districts of the State. HEV predisposing factors such as rural communities, agrarian communities, swine breeding communities, and proximity to water bodies were included in the selection criteria. A convenience sampling technique was used to recruit participants in the communities. All the apparently healthy participants per community that agreed to be a part of the study were recruited.

Ethical approval was obtained from the Health Planning. Research and Statistics Department of Osun State Ministry of Health with the approval number OSHREC/PRS/569T/52. Verbal consent was obtained from the community leaders and volunteers through advocacy and community sensitisation about the infection. Informed consent was obtained for participants under the age of 18 years from parents or guardians. All samples were collected from consenting individuals in these communities and stored at −80 °C prior analysis. The sample collection was performed between February 2015 and May 2017, while a structured questionnaire was used to collect demographic data and risk-associated factors from the participants. Serological and molecular assays were carried out on the samples at the Viral Gastroenteritis and Hepatitis Viruses and Enteroviruses unit (FG15) of the Robert Koch Institute, Berlin.

### 4.3. Anti-HEV IgM and IgG Assays

All samples were subjected to serological testing using the HEV ELISA 4.0 (MP Biomedicals Asia Pacific, Singapore) for anti-HEV total antibodies according to the manufacturer’s instructions. Samples that tested positive for anti-HEV total antibodies were further analysed for anti-HEV IgM using the HEV ELISA 3.0 (MP Biomedicals Asia Pacific, Singapore).

### 4.4. Viral RNA Extraction and RT-PCR

The presence of RNA was analysed in all sera samples which were tested positive for anti-HEV total antibodies and all faecal suspensions. Faecal suspensions were prepared by vortexing 0.1 g of faeces with 1 mL of phosphate-buffered saline. The suspensions were clarified at 10,000 rpm for 8 min at 25 °C [46]. RNA was extracted from a 140 μL mixture (135 μL serum/faecal suspension plus 5 μL MS2 phage-internal control) using the QIAmp Viral RNA kit (Qiagen, Hilden, Germany) and the QiaCUBE robotic machine according to the manufacturer’s instruction. Viral nucleic acids were eluted in 60 µL and stored at −80 °C until further use.

Two different PCR assays were used for the detection of RNA and the subsequent genotyping of PCR-positive samples: a one-step RT-nested PCR with primers located in the open reading frame 1 (ORF 1) region of the HEV genome and a one-step RT-semi-nested PCR assay with primers located in the ORF 2 region of the HEV genome (Table 4). Primers were designed to target conserved regions in multiple sequence alignments of the ORF 1 and ORF 2 regions of HEV-1 to HEV-4 [47]. The primers used were made available by the FG15 group of the Robert Koch Institute Berlin, Germany.

The ORF 1 and ORF 2 RT-PCRs were performed under the same conditions in a volume of 25 μL (5.8 µL water, 5 µL 5× RT-PCR buffer (2.5 mm Mg), 5 µL 5× Q-solution, 1 µL 10 mM dNTPs, 1 µL of antisense primer, 1 µL of sense primer, 0.2 µL RNasin (400 U/µL), 1 µL enzyme mix (RT/TaqPol) and 5 µL of template) using the One-Step RT-PCR Kit (Qiagen, Hilden, Germany). Cycling was performed in a Biometra Trio thermocycler (Biometra, Jena, Germany) under the following parameters for the first reaction: 50 °C for 30 min, 95 °C for 5 min followed by 10 cycles at 95 °C for 30 s, 60 °C (−1 °C/cycle) for 30 s, 72 °C for 45 s, and another 35 cycles at 95 °C for 30 s, 52 °C for 30 s, 72 °C for 45 s and a final extension at 72 °C for 7 min.

The second PCR amplification was performed in 25 µL volumes (10.5 µL water, 0.5 µL of antisense primer and 0.5 µL of sense primer, 12.5 µL Hot start Master Mix and 1 µL template- product from the first PCR) using the HotStarTaq master mix (Qiagen, Hilden, Germany).

The cycling parameters for the nested and semi-nested PCR were 95 °C for 10 min, followed by 40 cycles at 94 °C for 30 s, 52 °C for 30 s, 72 °C for 45 s, with a final extension at 72 °C for 5 min. A plasmid containing the HEV ORF 3 gene (HEV_RKI (GenBank accession no. FJ956757)) served as the positive control [47]. The sensitivity for the ORF 1 nested and ORF 2 semi-nested PCRs are 1 × 10^4^ copies/mL and 1 × 10^3^ copies/mL, respectively [47].

### 4.5. Statistical Analysis

The data were anonymously analysed descriptively and using logistic regression analyses (logistic command of STATA version 14.2. The descriptive statistics are presented as percentages. Univariable (adjusted by age and sex) and logistic regression was used to analyse the correlation of the risk factors obtained through the questionnaire with the laboratory findings. The level of significance was set at *p* < 0.05. To further confirm the risk factors associated with HEV seroprevalence, multivariate logistic regression analysis was carried out. For multivariate regression analysis we started with full model including all variables that were associated with HEV seroprevalence in the univariable analysis (*p* < 0.2). The final model was achieved by stepwise elimination of variables with the largest *p* and *p*> 0.2 untilall remaining variables were significantly associated with *p* < 0.05.

## 5. Conclusions

Conclusively, our study showed that seroprevalence of HEV is high in apparently healthy individuals from Osun State, Nigeria. Consequently Osun State can be termed endemic for HEV. This study also demonstrated that anti-HEV seropositivity increased with age, rurality, rearing of animals, and consumption of alcohol. Therefore, there is a need to prevent exposure to HEV infection by increasing accessibility to potable water, better farming practice, controlling spillover from animals and stepping-up good hygienic practices in the populace. More so, surveillance needs to be implemented and the burden of hepatitis E in Nigeria be assesed.

## Figures and Tables

**Figure 1 pathogens-09-00392-f001:**
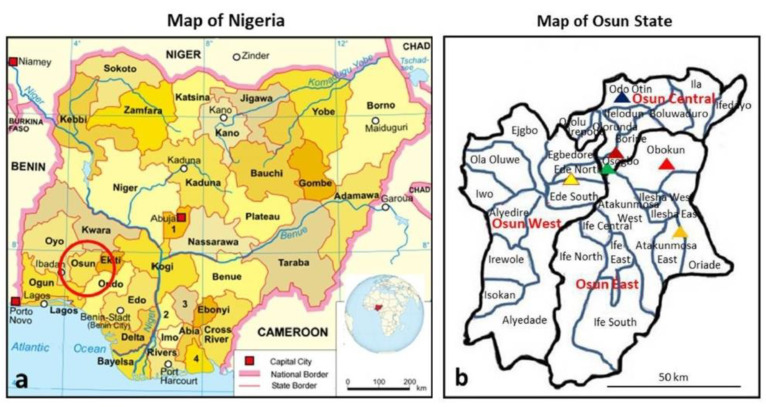
The sites where samples were collected: (**a**) map of Nigeria showing location of all States [44] (adapted from https://en.wikipedia.org/wiki/Free_Software_Foundation, https://en.wikipedia.org/wiki/Nigeria; inserted globe is from Nigeria on the globe (Africa centered).svg (creative commons licence CC BY-SA 3.0 and CC BY 4.0); (**b**) map of Osun State showing the three senatorial districts and the communities where samples were collected (map created by C.T.B. adapted from [45]). Note: each triangel denotes a site as Ede, Esa-Odo, Iperindo, Oke-Osun, Ore and Osogbo coloured with yellow, red, orange, brown, blue and green, respectively.

**Table 1 pathogens-09-00392-t001:** Seroprevalence of anti-Hepatitis E virus (HEV) total and IgM antibodies by sex and age.

			Anti-HEV Total Antibodies				Anti-HEV IgM			
		Number (%)	Positive (%)	aOR	95% CI	*p*	Positive (%)	aOR	95% CI	*p*
Sex	Female	356 (54.5)	45 (12.6)	Ref	−	−	10 (2.8)	Ref	−	−
	Male	297 (45.5)	53 (17.9)	1.5	1.0–2.4	0.061	15 (5.1)	1.9	0.8–4.5	0.127
Age * (years)	0–9	22 (3.4)	2 (9.1)	Ref	−	−	1 (4.6)	Ref	−	−
	10–19	68 (10.4)	7 (10.3)	1.2	0.2–6.1	0.86	2 (2.9)	0.7	0.1–7.5	0.73
	20–29	176 (27.0)	14 (8.0)	0.9	0.2–4.3	0.89	3 (1.7)	0.4	0.0–3.9	0.42
	30–39	117 (17.9)	10 (8.6)	1.0	0.2–4.9	0.99	2 (1.7)	0.4	0.0–4.6	0.46
	40–49	91 (13.9)	14 (15.4)	1.9	0.4–9.1	0.417	2 (2.2)	0.5	0.0–5.8	0.58
	50–59	73 (11.2)	23 (31.5)	4.9	1.1–22.9	**0.04**	4 (5.5)	1.3	0.1–12.5	0.81
	60–69	59 (9.0)	15 (25.4)	3.7	0.8–17.8	0.104	7 (11.9)	3.1	0.4–27.4	0.30
	>70	47 (7.2)	13 (27.7)	3.8	0.8–18.7	0.100	4 (8.5)	1.9	0.20–18.42	0.60
	Age **	653	98 (15.0)	1.03	1.0–1.0	**0.000**	25 (3.8)	1.03	1.0–1.1	**0.003**

Note: Ref = reference; aOR = adjusted odds ratio; CI = confidence interval; the CI are related to aOR, age * = categorical independent variable, age ** = continuous independent variable. Significant *p*-values are marked in bold.

**Table 2 pathogens-09-00392-t002:** Demographic and risk factors associated with HEV infection in apparently healthy members of the community.

Variables		Anti-HEV Total Antibodies				Anti-HEV IgM			
		Positive (%)	aOR	95% CI	*p*	Positive (%)	aOR	95% CI	*p*
	Ede	16 (8.0)	Ref			3 (1.5)	Ref		
	Esa-Odo	22 (22.0)	2.2	1.1–4.7	**0.04**	10 (10.0)	5.4	1.4–21.8	**0.017**
	Iperindo	15 (17.9)	1.8	0.8–3.9	0.15	0	1		
Location	Oke-Osun	20 (19.4)	2	1.0–4.2	0.06	7 (6.8)	3.7	0.9–14.9	0.07
	Ore	13 (21.0)	2	0.9–4.7	0.091	2 (3.2)	1.5	0.2–9.4	0.67
	Osogbo	12 (11.4)	1.3	0.6–2.9	0.492	3 (2.9)	1.7	0.3–8.8	0.51
Rurality	Urban	28 (9.2)	Ref	−	−	6 (2.0)	Ref	−	−
	Rural	70 (20.1)	1.8	1.1–3.0	**0.022**	19 (5.4)	2.1	0.8–5.6	0.15
	Married	71 (19.1)	Ref	−	−	18 (4.8)	Ref	−	−
Marital status	Divorced	0	1	−	−	0	1	−	−
	Single	15 (8.8)	1.1	0.5–2.5	0.766	4 (2.3)	1.3	0.3–5.9	0.71
	Widow	0	1	−	−	2 (100.0)	−	−	−
Lived abroad	Yes	4 (12.9)	0.5	0.2–1.7	0.28	1 (3.2)	0.4	0.1–3.5	0.44
	No	61 (16.0)	Ref	−	−	20 (7.1)	Ref	−	−
Alcohol consumption	Yes	21 (30.4)	2.4	1.3–4.4	**0.007**	7 (10.2)	2.8	1.0–7.7	**0.04**
	No	59 (13.8)	Ref	−	−	14 (3.0)	Ref	−	−
Jaundice exposure	Yes	7 (14.6)	0.8	0.3–01.8	0.532	2 (4.2)	0.9	0.2–4.1	0.9
	No	79 (15.8)	Ref	−	−	20 (4.0)	Ref	−	−
Religion	Christianity	22 (15.0)	Ref	−	−	2 (1.4)	Ref	−	−
	Muslim	11 (12.0)	0.7	0.3–1.6	0.381	2 (2.2)	1.6	0.2–11.6	0.65
	Traditional	0	1	−	−	0	1	−	−
Water source	Borehole	2 (5.0)	0.6	0.1–3.9	0.62	0	1	−	−
	Tap	21 (13.6)	1.4	0.2–9.4	0.76	5 (3.2)	0	0	−
	Sachet	2 (5.4)	0.5	0.1–3.3	0.47	0	1	−	−
	River	22 (22.9)	2	0.3–15.7	0.49	7 (7.3)	0	0	−
	Well	23 (18.1)	2	0.3–14.9	0.49	10 (7.9)	0	0	−
Water based activity	Yes	37 (18.0)	1.1	0.6–1.9	0.783	13 (6.3)	1.3	0.5–3.3	0.57
	No	28 (13.7)	Ref	−	−	8 (3.9)	Ref	−	−
Wash hands before eating	Yes	63 (15.3)	1	0.4–02.3	0.987	19 (4.6)	0.8	0.2–2.9	0.745
	No	8 (16.3)	Ref	−	−	3 (6.1)	Ref	−	−
Share of sharp objects	Yes	14 (15.9)	1.2	0.6–2.4	0.655	4 (4.6)	1.3	0.4–4.4	0.705
	No	30 (12.9)	Ref	−	−	8 (3.5)	Ref	−	−
Blood transfusion history	Yes	3 (7.9)	0.4	0.1–1.4	0.151	2 (5.3)	1.3	0.3–6.0	0.72
	No	83 (16.3)	Ref	−	−	20 (3.9)	Ref	−	−
Diabetic	Yes	2 (13.3)	0.7	0.2–3.5	0.698	0	1	−	−
	No	63 (15.9)	Ref			21 (5.1)	Ref		
On drugs	Yes	3 (11.1)	0.5	0.1–1.7	0.238	0	1	−	−
	No	61 (15.9)	Ref	−	−	21 (5.2)	Ref	−	−
Toilet type	Shot put	0 (0)	1	−	−	0	1	−	−
	Pit	58 (18.9)	1.6	0.8–3.2	0.152	19 (6.2)	2.1	0.6–7.6	0.25
	WC	13 (9.3)	Ref			3 (2.1)	Ref		
Rearing animals	Yes	57 (20.9)	3	1.3–6.7	**0.008**	19 (7.0)	3.5	0.8–16.2	0.1
	No	8 (5.8)	Ref	−		2 (1.4)	Ref	−	−
House type	F-to-F	5 (11.9)	Ref	−		0	1	−	−
	Flat	1 (11.1)	0.9	0.1–9.0	0.94	1 (11.1)	1	−	−

Note Ref = reference; aOR = adjusted odds ratio; CI = confidence interval; the CI are related to aOR, F-to-F = Face to face house. Significant *p*-values are marked in bold.

**Table 3 pathogens-09-00392-t003:** Multivariate logistic regression model evaluating odds ratios of risk factors associated with HEV seroprevalences.

Variables		Anti-HEV Total Antibodies							Anti-HEV IgM						
			Full model			Final Model				Full model			Final Model		
		Positive (%)	aOR	95% CI	*p*	aOR	95% CI	*p*	Positive (%)	aOR	95% CI	*p*	aOR	95% CI	*p*
	Ede	16 (8.0)	Ref						3 (1.5)	Ref			Ref		
	Esa-Odo	22 (22.0)	1.3	0.6–3.1	0.542	−	−	−	10 (10.0)	5.0	0.9–26.2	0.056	6.9	1.4–35.0	**0.020**
	Iperindo	15 (17.9)	−	−	−	−	−	−	0	−	−	−	1	−	−
Location	Oke-Osun	20 (19.4)	1.4	0.6–3.3	0.475	−	−	−	7 (6.8)	4.0	0.7–21.5	0.112	5.1	1.0–26.3	**0.051**
	Ore	13 (21.0)	1	−	−	−	−	−	2 (3.2)	1	−	−	1.3	0.2–10.1	0.801
	Osogbo	12 (11.4)	−	−	−	−	−	−	3 (2.9)	−	−	−	−	−	−
Rurality	Urban	28 (9.2)	Ref						6 (2.0)	Ref					
	Rural	70 (20.1)	1.6	0.4–6.0	0.484	−	−	−	19 (5.4)	1.0	0.1–8.8	0.968	−	−	
Alcohol consumption	Yes	21 (30.4)	2.8	1.4–5.7	**0.006**	2.7	1.3–5.3	**0.006**	7 (10.2)	3.8	1.3–11.1	0.016	3.8	1.3–11.2	**0.016**
	No	59 (13.8)	Ref			Ref			14 (3.0)	−	−	−	−	−	−
Blood transfusion history	Yes	3 (7.9)	0.8	0.2–2.9	0.686	−	−	−	2 (5.3)	−	−	−	−	−	−
	No	83 (16.3)	Ref						20 (3.9)	−	−	−	−	−	−
Toilet type	Shot put	0 (0)	1	−	−	−	−	−	0	−	−	−	−	−	−
	Pit	58 (18.9)	1	−	−	−	−	−	19 (6.2)	−	−	−	−	−	−
	WC	13 (9.3)	1.35	0.5–4.0	0.581	-	−	−	3 (2.1)	−	−	−	−	−	−
Rearing animals	Yes	57 (20.9)	2.4	0.9–6.5	**0.077**	3.2	1.4–7.3	**0.005**	19 (7.0)	1.9	0.3–11.7	0.514	−	−	−
	No	8 (5.8)	Ref						2 (1.4)				−	−	−

Note Ref = reference; aOR = adjusted odds ratio; CI = confidence interval; the CI are related to aOR. Significant *p*-values are marked in bold.

**Table 4 pathogens-09-00392-t004:** Primer Sequences.

	Sequence	Location *	Polarity
**ORF 1**			
HEV-38	5′-GAGGCYATGGTSGAGAARG-3′	4084–4102	+
HEV-39	5′-GCCATGTTCCAGACRGTRTTCC-3′	4622–4601	−
HEV-37	5′-GGTTCCGYGCTATTGARAARG-3′	4277–4297	+
HEV-27	5′-TCRCCAGAGTGYTTCTTCC-3′	4583–4565	−
**ORF 2**			
HEV-30	5′-CCGACAGAATTGATTTCGTCGG-3′	6296–6317	+
HEV-31	5′-GTCTTGGARTACTGCTGR-3′	6750–6733	−
HEV-32	5′-GTCTCAGCCAATGGCGAGCCRAC-3′	6350–6372	+

* Numbered according to HEV-1 (GenBank accession no. M73218).

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
