# Peer review of "Hepatitis E Virus Seroprevalence and Associated Risk Factors in Apparently Healthy Individuals from Osun State, Nigeria"

_pathogens, 2020, doi:10.3390/pathogens9050392_

Round 1

Reviewer 1 Report

The results of this study are not novel but previous studies were   based in small number of cases. This large cohort of study in healthy individuals Nigeria with well identify characteristics is very helpful. Collected samples include both plasma and feces, and both serology tests and HEV RNA were carried out. However, one of the major concerns about the manuscript is the low sensitivity of the HEV RNA used (10E3-10E4 copies/mL), fact that may possibly contribute to the absence of positive samples for HEV RNA. Adding  ALT levels is important and could to contribute to improve the manuscript

Minor comments

Please use low income countries instead of developing countries

Results

a multivariate analysis should be performed

Adding  ALT levels is important and could to contribute to improve the manuscript

Discussion

-A significant rate of subjects were antiHEV IgM positive, although none of them  tested positive for HEV RNA. The authors should comment on this fact.  Authors claimed that the rate is low, but in absence of positive HEV RNA it can be considerate high. Moreover,  some studies data have  showed long persistence of anti-HEV IgM, especially with some antiHEV IgM  tests (Riveiro-Barciela M, et al.  J Viral Hepat. 2020 Feb 27). Another explanation not mentioned by the authors is the possibility of false negative results from HEV RNA due to the low sensitivity of the used technique.

- Impact of age on seroprevalence also lies in the anti-HEV IgG used since its duration varies according to the test, with Wantai showing the longest duration.

Methods

- How and where was the selection of patients performed?

- Why HEV ARN was performed only in IgG-positive patients? It is well-know, especially important in blood donors, that during the window period patient can present detectable HEV RNA but negative serologies.

Table 1 and 2

The column of negative results (both IgG and IgM) can be obviated.

Author Response

Please see the attachment or you can view here:

Manuscript ID pathogens-795660

Osundare et al., “Hepatitis E Virus Seroprevalence and Associated Risk Factors in Apparently Healthy Individuals from Osun State, Nigeria”

Point-by-point response to the comments of the reviewers.

Author’s answers to the comments of the reviewers are highlighted in blue.

Line numbers and reference numbers mentioned in replies are based on the revised manuscript.

Changes in the manuscript are marked in yellow.

Reviewer comments and suggestions

Reviewer #1:

We are grateful to the reviewer for the time spent on the careful review of our manuscript.

  1. The results of this study are not novel but previous studies were  based in small number of cases. This large cohort of study in healthy individuals Nigeria with well identify characteristics is very helpful. Collected samples include both plasma and feces, and both serology tests and HEV RNA were carried out. However, one of the major concerns about the manuscript is the low sensitivity of the HEV RNA used (10E3-10E4 copies/mL), fact that may possibly contribute to the absence of positive samples for HEV RNA.

We sincerely thank the reviewer for this valuable comment; however, at the time of our study the sensitivity of the RT-PCR systems we used was approx. 10E+3 to 10E+4  copies/mL (840 to 8.4E+3 IU/ml) depending on the target region (ORF1 or ORF2) which equals to 25 and 250 copies/reaction, respectively. A recent report by Lhomme et al. (Lhomme et al. J Viral Hepat 2019, 26; 1139-1142) showed that in asymptomatic blood donors a median HEV load of 717 IU/ml was determined; however symptomatic individuals showed a median load of 2,82E+5 IU/ml. Therefore, in our opinion the PCR detection limit was acceptable for this study analysing apparently healthy individuals showing no HEV-like symptoms. We included a respective comment as a limitation of the study (line 283-286); however we cannot absolutely rule out a possible false-negative PCR result.

  1. Adding ALT levels is important and could to contribute to improve the manuscript

We agree with the reviewer that ALT levels are an important factor for hepatitis E. However, the study subjects were apparently healthy individuals and were therefore not tested on parameters of liver function like ALT levels. We unfortunately have no ALT data on hand.

Minor comments

  1. Please use low income countries instead of developing countries

Thank you for this comment. We have substituted “developing countries” with “low-income countries” throughout the text accordingly (lines 28, 54, 57 and 218).

Results

  1. a multivariate analysis should be performed

The multivariate logistic regression analysis has been performed now and has been included in the methodical description (line 380-385) and results section (lines 110-115 and new table 3). The inferences generated have also been added to the discussion and conclusion (lines 222-223, 242-243 and 269-270).

Discussion

  1. A significant rate of subjects were antiHEV IgM positive, although none of them tested positive for HEV RNA. The authors should comment on this fact.  Authors claimed that the rate is low, but in absence of positive HEV RNA it can be considerate high.

Thank you for pointing this out. “Low” has been reverted to “high” and a probable inference was also added in the text (lines 171 and 178-179)

  1. Moreover,  some studies data have  showed long persistence of anti-HEV IgM, especially with some antiHEV IgM  tests (Riveiro-Barciela M, et al.  J Viral Hepat. 2020 Feb 27). Another explanation not mentioned by the authors is the possibility of false negative results from HEV RNA due to the low sensitivity of the used technique.

We thank the reviewer for this valuable comment. We have added a respective comment to the text (lines 183-186;  “However, longer persistence of anti-HEV IgM (up till 34 months) have been reported with some assays and furthermore varying analytical performance of different assays has been shown, which should be taken into account in seroprevalence studies”).

  1. Impact of age on seroprevalence also lies in the anti-HEV IgG used since its duration varies according to the test, with Wantai showing the longest duration.

We are grateful to the reviewer for this comment. We have added a comment to this aspect and discussed it in the discussion section (lines 180-186).

 Methods

  1. How and where was the selection of patients performed?

The communities were chosen from the three Senatorial Districts of the State, after which convenience sampling technique was used to recruit the community dwellers in their communities. Apparently, healthy volunteers were recruited from the different communities in the States. Verbal consent was obtained from the community leaders and volunteers through advocacy and community sensitisation about the infection. Informed consent was obtained for participants under the age of 18 years from parents or guardians. All samples were collected from consenting individuals in these communities (lines 322-337).

  1. Why HEV RNA was performed only in IgG-positive patients? It is well-know, especially important in blood donors, that during the window period patient can present detectable HEV RNA but negative serologies.

The scope of the work did not include investigating samples that were in the window phase of infection. All the anti-HEV total antibodies samples were included to achieve a wider scope other than just the anti-HEV IgM positive samples. Additionally, all the stool samples were also negative for HEV RNA validating in part the results from blood.

  1. Table 1 and 2.The column of negative results (both IgG and IgM) can be obviated.

The negative columns in both tables have been deleted.

Reviewer 2 Report

Osundare et al. investigated in their manuscript 'Hepatitis E Virus Seroprevalence and Associated Risk Factors in Apparently Healthy Individuals from Osun State, Nigeria' the seroprevalence of hepatitis E virus (HEV) and associated risk factors in healthy individuals in Osun State, Nigeria. They analyzed serum and stool samples from healthy individuals in six communities from Osun State. HEV-specific antibodies (IgG and IgM) were analyzed by ELISA and HEV RNA by RT-PCR. Overall the authors could show that the seroprevalence is relatively equally distributed in all six communities. No individual was tested positive for HEV-RNA.

The overall topic of this manuscript is very interesting and important to understand hepatitis E virus infections in Nigeria. The manuscript in general is very well written.

All together I would suggest accepting the paper, if the authors would soften their interpretation that increasing sanitation would be the most important point to prevent HEV infection. The authors also used in their conclusion the term endemic, with is for a HEV seropositivity of 9% and 2% in urban and 20 % and 5% in rural regions (table 2, total HEV antibodies and IgM, respectively) a strong statement. Based on the data presented in this manuscript no data about the HEV genotype from the Osun State is available. Therefore, HEV infection can be caused by contaminated water, animals (housing pigs) or different lifestyle habits as meat consumption. Interestingly, as indicated in table 2 two-times more married people are HEV seropositive as singles. In summary, the compiled data are very nice and convincing, however I would suggest to soften the overall interpretation.

Reviewer 3 Report

Osundare and colleagues conducted a detailed seroprevalence study based on apparently healthy individuals from Nigeria. This report confirms previous findings on HEV seroprevalence from another study. The study is of sufficient detail and used state of the art diagnostic methods to determine anti-HEV IgM and IgG antibodies as well as HEV RNA in a subset of samples. To further improve the manuscript some comments:

• The recruitment of study participants should be stated in more detail. Volunteers, outpatients, convenience/left over samples from previous studies? How were they approached to participate?

• This should also be discussed in light of a possible sampling bias.

• The authors stated that all individuals were apparently healthy. This might be true at the time of recruitment but from Table 1 it becomes apparent that at least some were diabetic or received blood transfusion. This seems somehow contradictory. Please comment briefly.

• Technically, the group used an antibody assay from MP Diagnostic. Another widely used assay is the Wantai assay. Please comment on both assays in terms of sensitivity and specificity and comparability.

• The IgM results may indeed depict acute infections. However, false positives are frequently observed especially for IgM antibody detection methods. Please at least discuss this briefly.

• HIV is widespread in Nigeria and might affect serostatus. Do the authors have information on the HIV status of the study participants?
